# Projected resurgence of COVID-19 in the United States in July—December 2021 resulting from the increased transmissibility of the Delta variant and faltering vaccination

**Shaun Truelove[1]\*†, Claire P Smith[1]†, Michelle Qin[2], Luke C Mullany[1,3], Rebecca K Borchering[4], Justin Lessler[5], Katriona Shea[4], Emily Howerton[4], Lucie Contamin[6], John Levander[6], Jessica Kerr[6], Harry Hochheiser[6], Matt Kinsey[3], Kate Tallaksen[3], Shelby Wilson[3], Lauren Shin[3], Kaitlin Rainwater-Lovett[3], Joseph C Lemairtre[7], Juan Dent[1], Joshua Kaminsky[1], Elizabeth C Lee[1], Javier Perez-Saez[1], Alison Hill[1], Dean Karlen[8], Matteo Chinazzi[9], Jessica T Davis[9], Kunpeng Mu[9], Xinyue Xiong[9], Ana Pastore y Piontti[9], Alessandro Vespignani[9], Ajitesh Srivastava[10], Przemyslaw Porebski[11], Srinivasan Venkatramanan[11], Aniruddha Adiga[11], Bryan Lewis[11], Brian Klahn[11], Joseph Outten[11], Mark Orr[11], Galen Harrison[11], Benjamin Hurt[11], Jiangzhuo Chen[11], Anil Vullikanti[11], Madhav Marathe[11], Stefan Hoops[11], Parantapa Bhattacharya[11], Dustin Machi[11], Shi Chen[12], Rajib Paul[12], Daniel Janies[12], Jean-Claude Thill[12], Marta Galanti[13], Teresa K Yamana[13], Sen Pei[13], Jeffrey L Shaman[13], Jessica M Healy[14], Rachel B Slayton[14], Matthew Biggerstaff[14], Michael A Johansson[14], Michael C Runge[15]†, Cecile Viboud[16]†**

[1]Johns Hopkins Bloomberg School of Public Health, Johns Hopkins University, Baltimore, United States; [2]Harvard University, Cambridge, Massachusetts, United States; [3]Johns Hopkins University Applied Physics Laboratory, Laurel, United States; [4]Pennsylvania State University, University Park, United States; [5]University of North Carolina at Chapel Hill, Chapel Hill, United States; [6]University of Pittsburgh, Pittsburgh, United States; [7]École polytechnique fédérale de Lausanne, Lausanne, Switzerland; [8]University of Victoria, Victoria, Canada; [9]Northeastern University, Boston, United States; [10]University of Southern California, Los Angeles, United States; [11]University of Virginia, Charlottesville, United States; [12]University of North Carolina at Charlotte, Charlotte, United States; [13]Columbia University, New York, United States; [14]CDC COVID-19 Response Team, Atlanta, United States; [15]United States Geological Survey, Laurel, United States; [16]Fogarty International Center, National Institutes of Health, Bethesda, United States

**\*For correspondence:**
shauntruelove@jhu.edu

†These authors contributed equally to this work

**Abstract** In Spring 2021, the highly transmissible SARS-CoV-2 Delta variant began to cause increases in cases, hospitalizations, and deaths in parts of the United States. At the time, with slowed vaccination uptake, this novel variant was expected to increase the risk of pandemic resurgence in the US in summer and fall 2021. As part of the COVID-19 Scenario Modeling Hub, an ensemble of nine mechanistic models produced 6-month scenario projections for July–December 2021 for the United States. These projections estimated substantial resurgences of COVID-19 across

the US resulting from the more transmissible Delta variant, projected to occur across most of the US, coinciding with school and business reopening. The scenarios revealed that reaching higher vaccine coverage in July–December 2021 reduced the size and duration of the projected resurgence substantially, with the expected impacts was largely concentrated in a subset of states with lower vaccination coverage. Despite accurate projection of COVID-19 surges occurring and timing, the magnitude was substantially underestimated 2021 by the models compared with the of the reported cases, hospitalizations, and deaths occurring during July–December, highlighting the continued challenges to predict the evolving COVID-19 pandemic. Vaccination uptake remains critical to limiting transmission and disease, particularly in states with lower vaccination coverage. Higher vaccination goals at the onset of the surge of the new variant were estimated to avert over 1.5 million cases and 21,000 deaths, although may have had even greater impacts, considering the underestimated resurgence magnitude from the model.

## Editor's evaluation

In this paper, the authors presented the joint efforts of nine modeling teams to provide a six-month projection of the COVID-19 pandemic across the US, in view of the circulation of the more transmissible Delta variant. The results represented a timely assessment of the risk of COVID-19 resurgence in Summer 2021 when it was conducted in July 2021, and will be of historical interest as an example of modeling efforts to inform real-time decision making during the COVID-19 pandemic. This paper will be of high interest to public health specialists, forecast modelers, and members of the general public interested in the evolution of the COVID-19 pandemic and the impact of public health interventions in the USA.

## Introduction

The rapid development, scale-up, and deployment of COVID-19 vaccines in the United States (US) has been one of the biggest public health successes in the US during this pandemic, with reported cases in a nadir in June 2021 (*Centers for Disease Control and Prevention, 2021b*), despite increased testing capacities. With this success, non-pharmaceutical interventions (NPIs) were lifted, including mask mandates, in almost every jurisdiction across the US in Spring 2021. However, the emergence of novel variants with increased transmissibility and /or immune escape, particularly the Delta and Omicron variants, has continually raised concern about the potential timing and magnitude of the subsequent resurgence, and the ability to mitigate it through increased uptake of vaccination.

Established in December 2020, the COVID-19 Scenario Modeling Hub is an effort to apply a multiple-model approach to produce six-month projections of the state and national trajectories of cases, hospitalizations, and deaths in the US under defined scenarios (*Borchering et al., 2021*). Scenarios from projection rounds have focused on control measures, vaccination availability and uptake, emerging variants, and waning immunity (*COVID-19 Scenario Modeling Hub, 2020*). Projections are released in a timely manner to guide policy decisions and data are made publicly available on a website (*COVID-19 Scenario Modeling Hub, 2020*).

Here, we detail results from the seventh round of projections, in which increased transmissibility variants were incorporated into projections to assess the potential impact of the Delta variant. In all scenarios, resurgences across the US were projected, with the largest resurgences occurring for scenarios with the highest variant transmissibility (60% increase over the Alpha variant, which most closely resembles estimates for the Delta variant). Corresponding increases in hospitalizations and deaths were also projected. In scenarios with higher vaccine coverage, the size and duration of this resurgence was notably smaller. Cases were projected to increase in early July 2021 at the national level and peak in mid to late September 2021. Corresponding increases in hospitalizations and deaths were also projected. The resurgence was projected to be geographically heterogeneous; although most states were projected to experience some degree of rebound, those having higher vaccine coverage were projected to experience less severe increases in incidence relative to prior observed peaks. However, while the timing of these projected resurges was relatively accurate compared to what has since been reported, the magnitude of reported cases, hospitalization, and deaths far surpassed what was projected. Here, we describe our experience with multi-model projections of

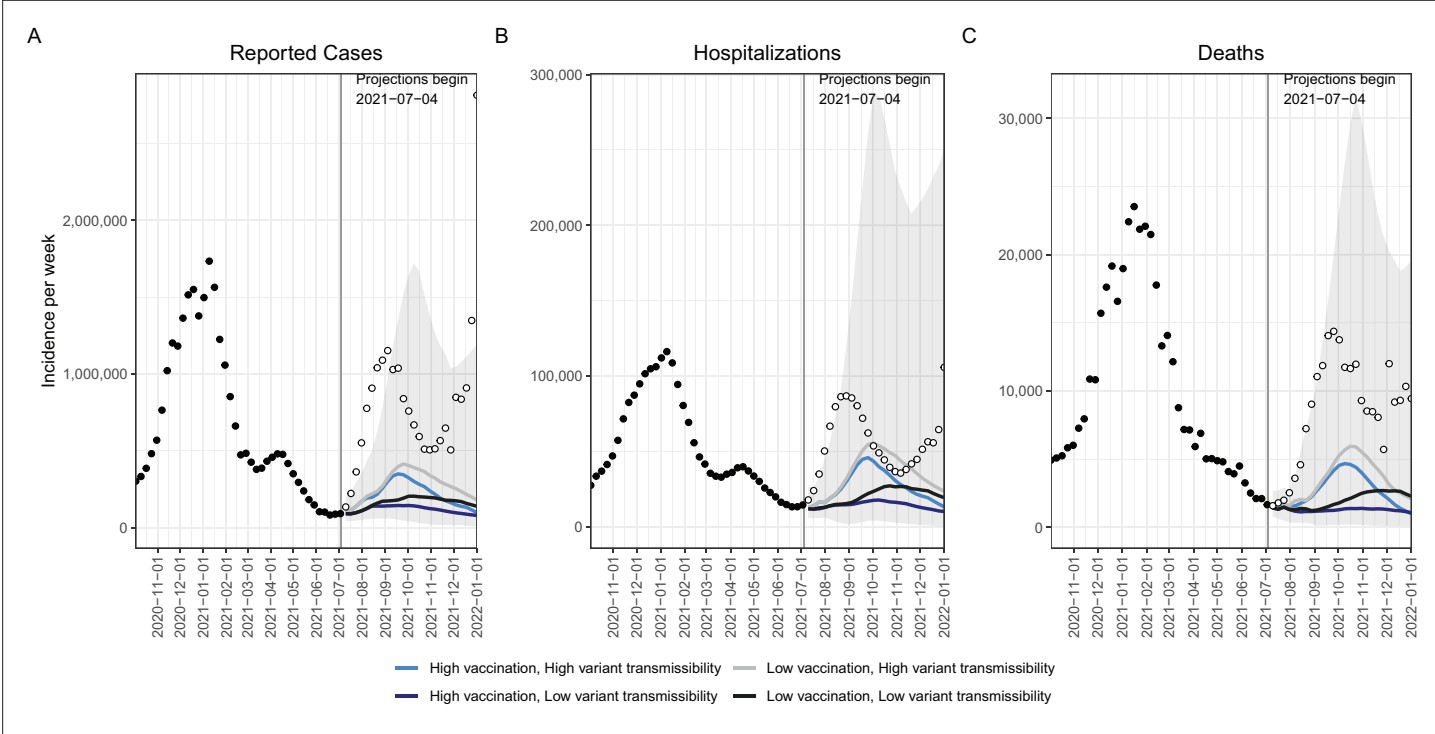

**Figure 1.** Historical data and weekly ensemble projections of reported numbers of COVID-19 cases. (**A**) Hospitalizations (**B**) and deaths (**C**) under four scenarios representing different levels of vaccination and Delta variant transmissibility increase — United States, October, 2020–December, 2021. Projections are ensemble estimates of 9 models projecting four 6-month scenarios with 95% prediction intervals (the grey shading encompasses the prediction intervals from all four scenarios). Projections used empirical data from up to July 3, 2021, to calibrate models (black filled dots). The vertical lines indicate the beginning of each projection, with only data available prior to that point used to fit the projections. Observations available after the projection start are displayed as open dots.

The online version of this article includes the following figure supplement(s) for figure 1:

**Figure supplement 1.** Historical data and weekly individual projections of reported numbers of COVID-19 cases under four scenarios representing different levels of vaccination and Delta variant transmissibility increase — United States, October 2020–December 2021.

**Figure supplement 2.** Historical data and weekly individual projections of reported numbers of COVID-19 hospitalizations under four scenarios representing different levels of vaccination and Delta variant transmissibility increase — United States, October 2020–December 2021.

**Figure supplement 3.** Historical data and weekly individual projections of reported numbers of COVID-19 deaths under four scenarios representing different levels of vaccination and Delta variant transmissibility increase — United States, October 2020–December 2021.

the Delta variant to highlight both the value of scenario-based projections for planning, but also the challenges to understand and predict the constantly evolving COVID-19 pandemic.

## Results

In the two scenarios with high Delta variant transmissibility (60% more transmissible than Alpha), we projected a national wave of cases to continue to grow over the summer and peak in mid- to late September 2021. In the scenario that assumes lower vaccination coverage among eligible individuals (70%) and higher variant transmissibility (the most pessimistic scenario), this resurgence was projected to peak at 414,000 weekly cases (95% projection interval (PI): 140,000–1,525,000) and 5900 weekly deaths (95% PI: 900–30,000) nationally. Overall, this scenario projected 7,554,000 (95% PI: 3,294,000–28,399,000) cumulative cases and 96,000 (95% PI: 27,000–476,000) cumulative deaths during July 4, 2021–Jan 1, 2022 (**Figure 1**).

With higher variant transmissibility, increasing national vaccination coverage was projected to temper the fall wave slightly and cause it to drop more quickly, but not prevent it. With an increase in national vaccine coverage to 80% by January 1, 2022, the ensemble projected 65,000 (16%) fewer cases and 1300 (21%) fewer deaths per week at the peak, and 1,525,000 (20%) and 21,000 (22%) fewer

cumulative cases and deaths, respectively, during July 4, 2021–January 1, 2022, when compared to the scenario where vaccination saturated at 70% nationally (*Figure 1*).

The projected national resurgence in COVID-19 cases in the higher transmissibility variant scenarios was composed of highly heterogeneous state-level resurgences. The ten states with the largest projected increases in incidence relative to their winter 2020–21 peak were, in descending order, Louisiana, Hawaii, Nevada, Arkansas, Florida, Missouri, Georgia, Alabama, Alaska, and Arizona. The ensemble estimates projected these states to experience median peak levels of weekly incident cases that were 18–69% (95% PI: 1%–541%) of their (smoothed) winter peak, although this was exceeded in many states. The 10 states with the lowest projected resurgences were, in ascending order, Massachusetts, Rhode Island, Vermont, Maine, Minnesota, Pennsylvania, North Dakota, Wisconsin, Tennessee, and South Dakota. The 10 states with the smallest projected resurgence had a median first-dose vaccine coverage of 70% among the eligible population (ages 12+) on July 3, 2021, compared to 56% in the ten states with the highest projected resurgence. We find a high negative correlation (Pearson's *r*=–0.66, *Figure 2*) between projected cumulative deaths per population and vaccination coverage on July 3, 2021 (*NCIRD, 2021*). In all states, even those with low overall vaccination coverage, at least 76% of people 65+had received at least one dose of the vaccine, which was expected to have a major effect in limiting mortality from Delta (*Centers for Disease Control and Prevention, 2021b*).

The impact of vaccination was already being observed early in the Delta wave: in the 10 states with the largest projected resurgence there was a 9% reduction in the observed case fatality ratio (CFR) comparing August–December 2020 and January–July 2021; in the 10 states with the smallest projected resurgence a 21% reduction in CFR was observed. During the projection period, we projected CFR reductions of 15% and 14%, as compared to August-December 2020. Lower transmissibility variant scenarios projected significantly reduced resurgence, projecting cumulative national cases of only 9% and 13% compared to the winter 2020–21 peak. Similarly, in the previous projection round (Round 6), which similar to the 7th round except it assumed only a 20% transmissibility increase from a novel variant, resurgence was expected to produce only 8% of the cases reported during the winter 2020–21 peak nationally (*COVID-19 Scenario Modeling Hub, 2020*).

Weekly case observations exceeded our 95% projection interval in the first 9 weeks of the projection period in all scenarios, including those assuming 60% increased transmissibility of the Delta variant (*Figure 1*). Our projections also tended to underestimate hospitalizations and deaths in the ascending phase of the Delta wave, although to a lesser degree. We compared weekly incident and cumulative cases during the first four weeks after the projection date (July 4–31, 2021). The total median projected number of cases underestimated the observed cases overall during this 4-week period (1,256,000 observed vs 516,000 projected); however, we find a strong correlation between ranking of observed and projected total cases per 100,000 during the first four weeks of the projection period, at the state level (Spearman's ρ=0.87, *Figure 3*). Seven of the ten states with greatest projected incidence rank in the ten worst observed incidence states. Hence while projections did not capture the full scope of the rise in incidence due to the Delta variant, these projections reflected the expected severity ranking among state projections well.

The two high variant transmissibility scenarios, which most closely resembled the characteristics of the circulating Delta variant, projected the timing of the Delta resurgence, projecting deaths to increase simultaneously with reported cases and peak one week after reported cases (*Figure 1*). However, all scenarios substantially underestimated the magnitude of the Delta wave for all outcomes. Among the two high variant transmissibility scenarios, at the national level the peak cases were under-projected by 70% (95% PI: −25–91%) and 64% (95% PI: −49–88%) (349,183 and 413,733 vs 1.15 M peak reported weekly cases), though the 95% projection interval did capture the reported magnitude of the peak (*Figure 1*). Similarly, hospitalizations and deaths were also under-projected by 47% (95% PI: −184–84%) and 36% (95% PI: −230–82%) (46,000 and 56,000 vs 87,000 peak reported hospitalizations) and 67% (95% PI: −77–93%) and 59% (95% PI: −120–93%) (4700 and 6000 vs 14,000 peak reported deaths), respectively; both also captured the reported magnitude within projection intervals.

## Discussion

Prevalence of the SARS-CoV-2 Delta variant rose quickly in the US between May and June 2021, with the variant achieving dominance by late June 2021, and accounting for over 90% of all SARS-CoV-2 infections for an extended period from late July to early December 2021 (*Centers for Disease*

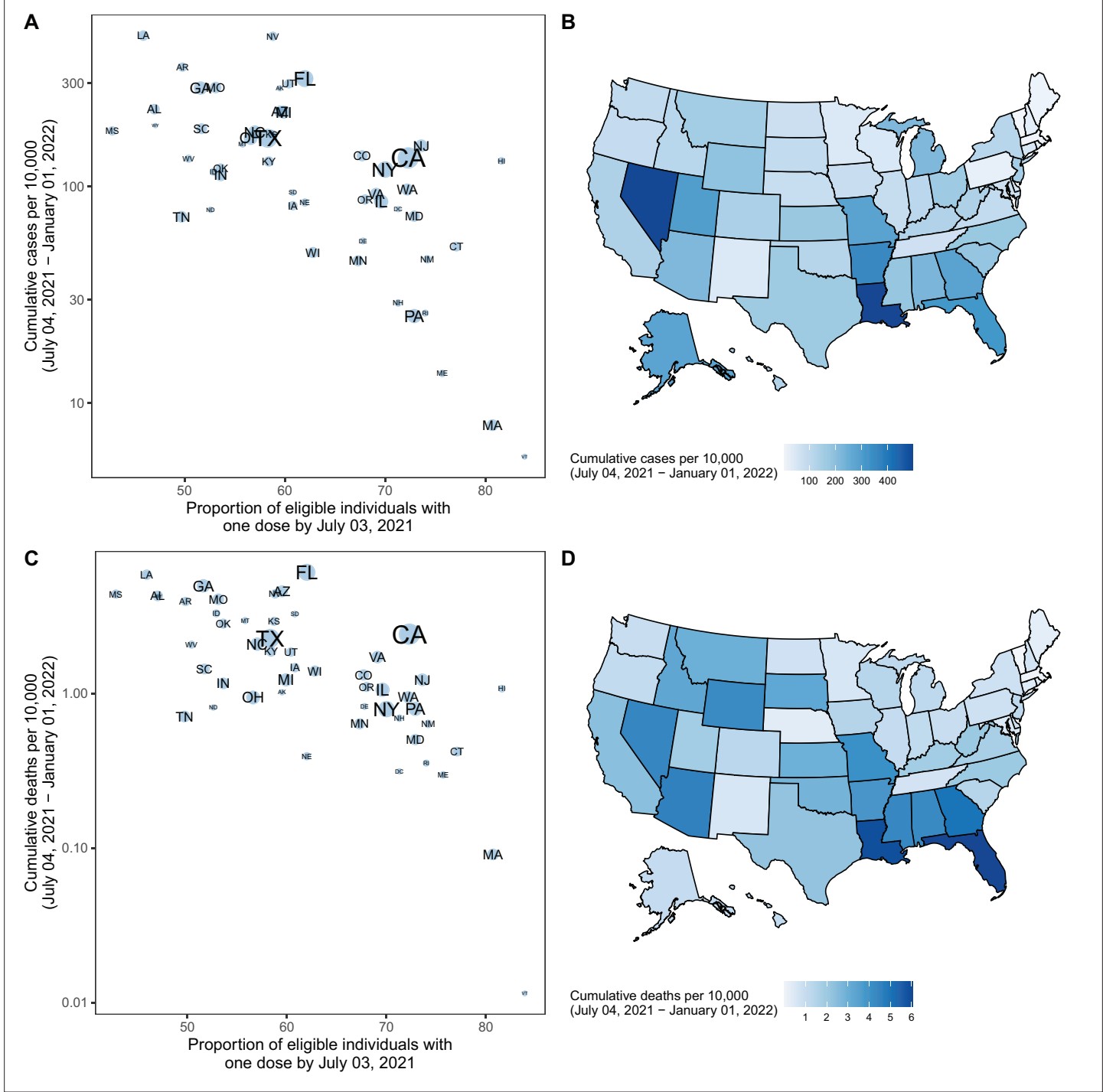

**Figure 2.** Projected cumulative cases and mortality in the most pessimistic scenario (low vaccination, high variant transmissibility) and current vaccination coverage by state — United States, July 4, 2021–January 1, 2022. (**A**) Correlation between cumulative projected cases per 10,000 population during the 6-month period and proportion of the eligible population vaccinated with at least one COVID-19 vaccine dose by July 3, 2021, by state. Circle sizes represent population size. Single dose coverage was used as data reporting were most reliable for the first dose at the time of this analysis; yet second dose coverage is highly correlated with first dose coverage (Pearson rho = 0.92 on July 3, 2021, p<10$^{-15}$). (**B**) Cumulative projected cases per 10,000 population during the 6-month period, by state. (**C**) Correlation between cumulative projected deaths per 10,000 population during the 6-month period and proportion of the eligible population vaccinated with at least one COVID-19 vaccine dose by July 3, 2021, by state. Circle sizes represent population size. (**D**) Cumulative projected deaths per 10,000 population during the 6-month period, by state.

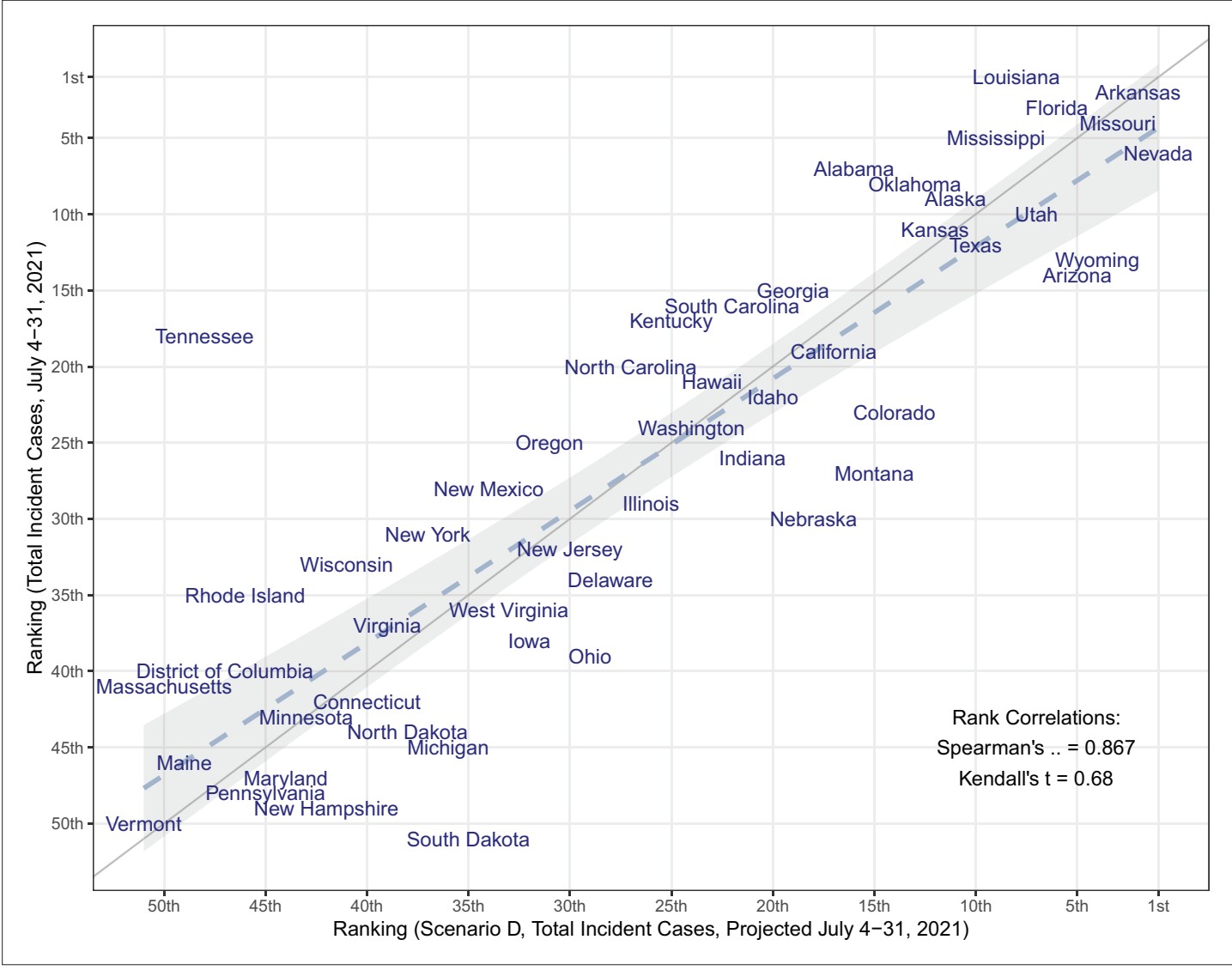

**Figure 3.** Comparison of the median projected and observed state-level total COVID-19 case incidences occurring during July 4–31, 2021, United States. Comparison is based on ranking of incidence per capita in 50 states + DC (Spearman's rank correlation = 0.867). The grey solid line represents perfect agreement between ranks (y=x), which overlays a regression line fitted to the data (dashed line) and 95% confidence intervals (grey shaded area).

*Control and Prevention, 2021b*). This variant prompted concerns about the scale of the COVID-19 resurgence in the US in the summer and fall of 2021, especially in the midst of decreased NPIs and slowing vaccination rates. Projections combining insights from multiple models suggested sizable resurgences of COVID-19 across the US, assuming growth of a variant that is 60% more transmissible than the Alpha variant (an assumption aligned with most estimates of the relative transmissibility of the Delta variant) (*Allen et al., 2022*). In scenarios with higher vaccination coverage, the magnitude of the resurgence in cases and deaths was substantially lower than in the lower coverage scenarios. Efforts to increase vaccination rates are critical and will save lives before and during future resurgences. At the outset of the Delta wave on July 1, 2021, only 13 states and Washington, D.C. had accomplished President Biden's goal for vaccination coverage among eligible populations at or above 70%.

The rapid case growth observed in July 2021 in multiple US states was surprising, tracking with or above the projections from our worst-case scenario. Scenarios were designed at the end of June 2021 based on information available at that time about the transmissibility of the Delta variant, vaccine effectiveness, and vaccine coverage; projections were generated on or before July 4, 2021. Our data should not be understood as forecasts, but as projections conditional on the scenarios

and assumptions; several reasons could explain why case growth was faster than expected, including key epidemiological and behavioral aspects that may have affected the disease dynamics. Possible mechanisms driving the underestimation of the observed Summer 2021 resurgence may include inaccurate assumptions about the transmissibility and severity of Delta (including changes in serial interval or severity of infection relative to other variants), the effectiveness of the vaccines against infection and transmission, waning of natural and vaccine-derived immunity, changes in testing practices, and the interaction of these factors with NPIs and behavior change (*Li et al., 2021*; *Elliott et al., 2021*; *Puranik et al., 2021*). Assumptions regarding these factors were left to the discretion of the teams and therefore vary across models. Critically, eight of the nine teams did not include waning of natural or vaccine-derived immunity, an assumption that we now know is incorrect. (*Higdon et al., 2021*) Subsequent rounds of projection have focused on different scenarios of waning immunity. Yet in this round, absence of waning resulted in a much smaller population susceptible to infection with Delta, thus reducing the overall potential magnitude that could be projected by these models. Use of NPIs has been substantially reduced across the US, with a lapse in mask mandates in most states. Although modeled use of NPIs was left to the discretion of individual modeling teams and did not vary between scenarios, results of prior rounds underscore the effectiveness of NPIs, in combination with increasing vaccination, to moderate the spread of a highly transmissible variant (*Borchering et al., 2021*). The extent to which NPIs may be necessary will vary across states, as those states with high levels of vaccination coverage or natural immunity may be at lower risk for an increase in cases.

The impact of the resurgence on severe disease and mortality was expected to vary substantially across states; states with younger populations and higher vaccination coverage among older and high-risk populations were expected to experience a relatively lower burden of severe disease, even with resurgences in cases. Several states (for example, South Dakota, North Dakota) with low vaccination coverage were not projected to experience major resurgences, likely because of high naturally acquired immunity. In addition, as the projected resurgence continued throughout the summer and into the beginning of the school year, efforts to promote vaccination among eligible school-aged children and college students and to maintain key prevention strategies in schools (for example, mask-wearing among the unvaccinated, physical distancing, screening programs) likely helped reduce risks with a safe return to in-person instruction (*Centers for Disease Control and Prevention, 2021a*). Observed increases in new vaccinations, particularly among young age groups and in jurisdictions most severely impacted by the Delta variant, was a positive step in this direction (*Centers for Disease Control and Prevention, 2021b*).

The findings in this report are subject to several limitations. First, considerable uncertainty is inherent to long-term projections. This has been repeatedly illustrated throughout the COVID-19 pandemic, with rapid changes in behavior, deployment of vaccines and boosters, and the emergence of novel variants, each of which has the capacity to drastically shift the epidemic trajectories. Uncertainty may arise from three main sources: specification of the scenarios (for example, uncertainty in transmissibility); errors in the structure or assumptions of individual models given a specific scenario (for example, variations in assumptions about vaccination uptake); and inaccurate calibration based on incomplete or biased data (for example, reporting backlogs). None of the four scenarios considered here were likely to precisely reflect the future reality over a 6-month period. As a case in point the emergence of the Omicron variant in December 2021, at the end of our projection period, could not have been predicted when scenarios were designed in June 2021 (*Borchering et al., 2021*). Similarly, a resurgence in Delta variant incidence was observed in mid-fall 2021, possibly due to changes in behavior and waning immunity, and is not captured in scenarios or model projections. Further, for a given scenario, there is notable variation among individual model projections with regard to both the timing and the magnitude of the resurgence (*Figure 1—figure supplements 1–3*). Variation likely reflects differences in model structure, projected vaccine coverage, projected variant growth, and importance of seasonal effects. Some of these variations reflect true scientific uncertainty, making ensemble projections particularly useful to integrate uncertainty between and within individual models. In addition, these scenarios do not specify considerations of Delta infecting previously immune individuals due to moderate antigenic changes, the waning of existing immunity, increases in NPIs, or vaccination among children aged <12 years starting in November 2021, all of which were expected to be important drivers of dynamics in the subsequent months. In the same vein, model estimates are dependent on assumptions about vaccine hesitancy, which are informed in

part by large-scale surveys of vaccine sentiments (*Carnegie Mellon University Delphi Group, 2021*; *Estimates of Vaccine Hesitancy for COVID-19, 2021*). These surveys may underestimate vaccine hesitancy, as coverage estimates among survey respondents are substantially higher than measured among the overall US population. Additionally, there are limitations to individual component models, although these concerns are tempered by analyzing ensembles of the nine different models. Overall, a full evaluation of our projections and sources of uncertainty is particularly difficult in a scenario context and is beyond the scope of this paper. However, it is worth noting that in this particular round of projection, the relationship between projection accuracy and time horizon is not straightforward (e.g. refer to *Figure 1* for a visual assessment of coverage).

## Conclusions

The emergence and introduction of more transmissible SARS-CoV-2 variants like the Delta variant was projected to lead to a substantial resurgence of COVID-19 in the US, which was observed in every state across the country. The high variant transmissibility scenarios, which more accurately represented the characteristics of the Delta variant, both in transmissibility and in current case trajectories, projected a significant national resurgence with substantial variation in magnitude across states. Resurgences were expected to be more pronounced in low-vaccination jurisdictions. The projections indicated that even with substantial vaccination coverage, the increased transmissibility of new variants like Delta can continue to challenge our ability to control this pandemic. Renewed efforts to increase vaccination coverage are critical to limiting transmission and disease, particularly in states with low natural immunity and lower current vaccination, in addition to re-instituting control measures like indoor masking when needed. Projections of Delta resurgence presented in this paper were made publicly available in early July 2021 (*COVID-19 Scenario Modeling Hub, 2020*), 2 months ahead of the peak of the Delta wave, providing actionable results. There is a trade-off between releasing projections in a timely manner to guide decisions, and projection accuracy and uncertainty that improve with incorporation of recent information. While these projections dramatically underestimated the magnitude of the Delta resurgence, demonstrating the challenges to predict this continually evolving pandemic, they did provide value in projecting the timing and emphasizing the importance of vaccination. Multi-model ensemble efforts such as the COVID-19 Scenario Modeling Hub are particularly well-suited to provide disease projections to inform the pandemic response under changing epidemiological and behavioral situations.

## Materials and methods

The COVID-19 Scenario Modeling Hub (*COVID-19 Scenario Modeling Hub, 2020*) convened nine modeling teams in an open call to provide six-month (July 3, 2021-January 1, 2022) COVID-19 projections in the US using data available through July 3, 2021. Each team developed a model to project weekly reported cases, hospitalizations, and deaths, both nationally and by jurisdiction (50 states and the District of Columbia), for four different epidemiological scenarios. Models were calibrated against data from the Johns Hopkins Center for Systems Science and Engineering Coronavirus Resource Center and federal databases (*Coronavirus Resource Center, 2020*; *US Department of Health and Human Services, 2020*). The four scenarios included low and high vaccination hesitancy levels, assuming national vaccination coverage saturation at 80% and 70%, respectively, based on hesitancy surveys (*Table 1*) (*Carnegie Mellon University Delphi Group, 2021*; *Estimates of Vaccine Hesitancy for COVID-19, 2021*). Participating teams accounted for vaccination rates by state, age, and risk-groups (for example, older adults and health care workers). Specified vaccine efficacy levels were constant across the scenarios and were based on protection against clinical disease in randomized clinical trials and effectiveness studies; parameters for effectiveness against infection, transmission, and progression to severe outcomes (for example, death) were left to be specified by each team (*COVID-19 Scenario Modeling Hub, 2020*). When the scenarios were designed in late June 2021, little information was available on vaccine efficacy specific to the Delta variant and on waning immunity. For details on individual model assumptions, see *Supplementary file 1*.

Scenarios assumed one of two levels of increased transmissibility for the Delta variant: 40% (low) or 60% (high) more transmissible than the Alpha variant. Increases in new variant prevalence over time were determined by each modeling team and were estimated at the state level.

**Table 1.** COVID-19 projection scenarios* — United States, July 4, 2021–January 1, 2022.
Scenarios defined for projection of COVID-19 cases, hospitalizations, and deaths for the sixth round of projections through the COVID-19 Scenario Modeling Hub§.

| | Low impact variant; (*low transmissibility increase*) | High impact variant; (*high transmissibility increase*) |
|---|---|---|
| **High vaccination;** (*low hesitancy*) | *Vaccination:*<br>• Coverage saturates at 80% nationally among the vaccine-eligible population* by December 31, 2021†<br>• VE is 50%/90% for Pfizer/Moderna against currently circulating variants (1st /2nd dose) and 60% for J&J (1 dose)<br>• J&J no longer used†*<br><br>*Variant:*<br>• 40% increased transmissibility as compared with Alpha for Delta variant. Initial prevalence estimated at state-level by teams. | *Vaccination:*<br>• Coverage saturates at 80% nationally among the vaccine-eligible population* by December 31, 2021†<br>• VE is 35%/85% for Pfizer/Moderna against currently circulating variants (1st /2nd dose) and 60% for J&J (1 dose)<br>• J&J no longer used†*<br><br>*Variant:*<br>• 60% increased transmissibility as compared with Alpha for Delta variant. Initial prevalence estimated at state-level by teams. |
| **Low vaccination;** (*high hesitancy*) | *Vaccination:*<br>• Coverage saturates at 70% nationally among the vaccine-eligible population* by December 31, 2021†<br>• VE is 50%/90% for Pfizer/Moderna against currently circulating variants (1st /2nd dose) and 60% for J&J (1 dose)<br>• J&J no longer used†*<br><br>*Variant:*<br>• 40% increased transmissibility as compared with Alpha for Delta variant. Initial prevalence estimated at state-level by teams. | *Vaccination:*<br>• Coverage saturates at 70% nationally among the vaccine-eligible population* by December 31, 2021†<br>• VE is 35%/85% for Pfizer/Moderna against currently circulating variants (1st /2nd dose) and 60% for J&J (1 dose)<br>• J&J no longer used ‡<br><br>*Variant:*<br>• 60% increased transmissibility as compared with Alpha for Delta variant. Initial prevalence estimated at state-level by teams. |

*The Vaccine-eligible population is presumed to be individuals aged 12 years and older through the end of the projection period.

†Vaccine hesitancy expected to cause vaccination coverage to slow and eventually saturate at some level below 100%. The saturation levels provided in these scenarios are National reference points to guide defining hesitancy, though the speed of that saturation and heterogeneity between states (or other geospatial scales) and/or age groups are at the discretion of the modeling team (**COVID-19 Scenario Modeling Hub, 2020**). The high vaccination 80% saturation is defined using the current estimates from the Delphi group (updated from Round 6) (**Carnegie Mellon University Delphi Group, 2021**). The low saturation estimate of 70% is the lowest county-level estimate from the US Census Bureau's Pulse Survey from May 26-June 7, 2021 data (**Estimates of Vaccine Hesitancy for COVID-19, 2021**).

‡To simplify the models and future projections of vaccine administration, it was assumed continued administration of the Johnson & Johnson (J&J) vaccine would not occur on or after the projection date (after July 4, 2021) due to the limited amount administered previously in the US (as of August 4, 2021 approximately 4 million doses delivered since April 13, 2021 compared to 153 million for Pfizer and Moderna) (**Centers for Disease Control and Prevention, 2021b**).

§COVID-19 Scenario Modeling Hub: https://covid19scenariomodelinghub.org/.

Individual models differed substantially in structure and design; see *Supplementary file 1* and the COVID-19 Scenario Modeling Hub GitHub website for more details (**COVID-19 Scenario Modeling Hub, 2021**). Individual modeling teams provided probabilistic projections of incident and cumulative epidemic trajectory for each week of the projection period, with 23 quantiles requested (0.01, 0.025, 0.05, every 5%–0.95, 0.975, and 0.99). These individual projections were combined into an ensemble for each scenario, outcome, week, and location using an equally-weighted linear opinion pool method across teams that trimmed the highest and lowest model at each point and quantile (**Stone, 1961**; **Jose et al., 2014**). Point estimates provided here are the median of the ensemble.

For any given pair of scenarios, averted cases and deaths were calculated as the difference (and ratio) between the median point estimates of the ensemble for the two scenarios. To provide a relative measure of resurgence in each state, we compared the intensity of the projected outbreak in the next six months to the size of the winter 2020–2021 outbreak – a period of high hospital burden in many jurisdictions. Specifically, projected resurgences were assessed by taking the ratio of the peak projected median incidence in a given location over the projection period (July 3, 2021-January 1, 2022) to the highest incidence experienced during the winter 2020–2021 period (defined as October 1, 2020–February 28, 2021) for the same location. Winter 2020–21 peaks were identified as the seven-day average centered around the day with the highest incident cases from smoothed curves generated through a penalized cubic spline Poisson regression model fit to the incident cases.

Details on the data used by each model can be found in *Supplementary file 1*, with further details found on the COVID-19 Scenario Modeling Hub GitHub repository website (https://github.com/midas-network/covid19-scenario-modeling-hub; DOI: 10.5281/zenodo.6584489) (*COVID-19 Scenario Modeling Hub, 2021*). All model output data and ensembled estimates are publicly available on the GitHub repository. All code used to generate numbers and figures reported in this manuscript are publicly available via the GitHub repository. Code required for ensembling model outputs can be made available upon request. Figure, code, and data are available through the open-source MIT license.

Disclaimer: The findings and conclusions in this report are those of the authors and do not necessarily represent the views of the Centers for Disease Control and Prevention or the National Institutes of Health. Any use of trade, firm, or product names is for descriptive purposes only and does not imply endorsement by the US Government.

## Acknowledgements

IHME (Bobby Reiner) for helpful discussions.

## Additional information

### Competing interests

Justin Lessler: has served as an expert witness on cases where the likely length of the pandemic was of issue. Jeffrey L Shaman: and Columbia University disclose partial ownership of SK Analytics. Discloses consulting for BNI. Michael C Runge: reports stock ownership in Becton Dickinson & Co, which manufactures medical equipment used in COVID testing, vaccination, and treatment. The other authors declare that no competing interests exist.

### Funding

| Funder | Grant reference number | Author |
|---|---|---|
| National Science Foundation | 2127976 | Shaun Truelove<br>Claire P Smith<br>Juan Dent<br>Joshua Kaminsky<br>Elizabeth C Lee<br>Alison Hill |
| National Science Foundation | 2028301 | Rebecca K Borchering<br>Katriona Shea |
| Huck Institutes of the Life Sciences | | Katriona Shea<br>Emily Howerton |
| National Institute of General Medical Sciences | 5U24GM132013-02 | Lucie Contamin<br>John Levander<br>Jessica Kerr<br>Harry Hochheiser |
| United States Department of Health and Human Services | 75A50121C00003 | Luke C Mullany<br>Matt Kinsey<br>Kate Tallaksen<br>Shelby Wilson<br>Lauren Shin<br>Kaitlin Rainwater-Lovett |
| United States Department of Health and Human Services | 6U01IP001137 | Jessica T Davis<br>Ana Pastore y Piontti<br>Alessandro Vespignani |
| United States Department of Health and Human Services | 5U01IP0001137 | Matteo Chinazzi<br>Kunpeng Mu<br>Xinyue Xiong<br>Alessandro Vespignani |

| Funder | Grant reference number | Author |
|---|---|---|
| National Science Foundation | 2027007 | Ajitesh Srivastava |
| National Science Foundation | 2126278 | Rebecca K Borchering<br>Katriona Shea |
| United States Department of Health and Human Services | | Shaun Truelove<br>Claire P Smith<br>Justin Lessler<br>Juan Dent<br>Joshua Kaminsky<br>Elizabeth C Lee<br>Javier Perez-Saez<br>Alison Hill |
| California Department of Public Health | | Shaun Truelove<br>Claire P Smith<br>Justin Lessler<br>Juan Dent<br>Joshua Kaminsky<br>Elizabeth C Lee<br>Javier Perez-Saez |
| Johns Hopkins University | | Shaun Truelove<br>Claire P Smith<br>Justin Lessler<br>Juan Dent<br>Joshua Kaminsky<br>Elizabeth C Lee<br>Javier Perez-Saez<br>Alison Hill |
| National Institutes of Health | R01GM140564 | Justin Lessler |
| Swiss National Science Foundation | 200021--172578) | Joseph C Lemairtre |
| National Institutes of Health | R01GM109718 | Przemyslaw Porebski<br>Srinivasan Venkatramanan<br>Aniruddha Adiga<br>Bryan Lewis<br>Brian Klahn<br>Joseph Outten<br>Mark Orr<br>Galen Harrison<br>Benjamin Hurt<br>Jiangzhuo Chen<br>Anil Vullikanti<br>Madhav Marathe<br>Stefan Hoops<br>Parantapa Bhattacharya<br>Dustin Machi |
| Virginia Department of Health | VDH-21-501-0135 | Przemyslaw Porebski<br>Srinivasan Venkatramanan<br>Aniruddha Adiga<br>Bryan Lewis<br>Brian Klahn<br>Joseph Outten<br>Mark Orr<br>Galen Harrison<br>Benjamin Hurt<br>Jiangzhuo Chen<br>Anil Vullikanti<br>Madhav Marathe<br>Stefan Hoops<br>Parantapa Bhattacharya<br>Dustin Machi |

| Funder | Grant reference number | Author |
|---|---|---|
| National Science Foundation | OAC-1916805 | Przemyslaw Porebski<br>Srinivasan Venkatramanan<br>Aniruddha Adiga<br>Bryan Lewis<br>Brian Klahn<br>Joseph Outten<br>Mark Orr<br>Galen Harrison<br>Benjamin Hurt<br>Jiangzhuo Chen<br>Anil Vullikanti<br>Madhav Marathe<br>Stefan Hoops<br>Parantapa Bhattacharya<br>Dustin Machi |
| National Science Foundation | CCF-1918656 | Przemyslaw Porebski<br>Srinivasan Venkatramanan<br>Aniruddha Adiga<br>Bryan Lewis<br>Brian Klahn<br>Joseph Outten<br>Mark Orr<br>Galen Harrison<br>Benjamin Hurt<br>Jiangzhuo Chen<br>Anil Vullikanti<br>Madhav Marathe<br>Stefan Hoops<br>Parantapa Bhattacharya<br>Dustin Machi |
| National Science Foundation | CCF-2142997 | Przemyslaw Porebski<br>Srinivasan Venkatramanan<br>Aniruddha Adiga<br>Bryan Lewis<br>Brian Klahn<br>Joseph Outten<br>Mark Orr<br>Galen Harrison<br>Benjamin Hurt<br>Jiangzhuo Chen<br>Anil Vullikanti<br>Madhav Marathe<br>Stefan Hoops<br>Parantapa Bhattacharya<br>Dustin Machi |
| National Science Foundation | OAC-2027541 | Przemyslaw Porebski<br>Srinivasan Venkatramanan<br>Aniruddha Adiga<br>Bryan Lewis<br>Brian Klahn<br>Joseph Outten<br>Mark Orr<br>Galen Harrison<br>Benjamin Hurt<br>Jiangzhuo Chen<br>Anil Vullikanti<br>Madhav Marathe<br>Stefan Hoops<br>Parantapa Bhattacharya<br>Dustin Machi |

| Funder | Grant reference number | Author |
| --- | --- | --- |
| National Science Foundation | TG-BIO210084 | Przemyslaw Porebski<br>Srinivasan Venkatramanan<br>Aniruddha Adiga<br>Bryan Lewis<br>Brian Klahn<br>Joseph Outten<br>Mark Orr<br>Galen Harrison<br>Benjamin Hurt<br>Jiangzhuo Chen<br>Anil Vullikanti<br>Madhav Marathe<br>Stefan Hoops<br>Parantapa Bhattacharya<br>Dustin Machi |
| Centers for Disease Control and Prevention | 75D30119C05935 | Przemyslaw Porebski<br>Srinivasan Venkatramanan<br>Aniruddha Adiga<br>Bryan Lewis<br>Brian Klahn<br>Joseph Outten<br>Mark Orr<br>Galen Harrison<br>Benjamin Hurt<br>Jiangzhuo Chen<br>Anil Vullikanti<br>Madhav Marathe<br>Stefan Hoops<br>Parantapa Bhattacharya<br>Dustin Machi |
| Defense Threat Reduction Agency | S-D00189-15-TO-01-UVA | Przemyslaw Porebski<br>Srinivasan Venkatramanan<br>Aniruddha Adiga<br>Bryan Lewis<br>Brian Klahn<br>Joseph Outten<br>Mark Orr<br>Galen Harrison<br>Benjamin Hurt<br>Jiangzhuo Chen<br>Anil Vullikanti<br>Madhav Marathe<br>Stefan Hoops<br>Parantapa Bhattacharya<br>Dustin Machi |
| Centers for Disease Control and Prevention | 200-2016-91781 | Shaun Truelove<br>Claire P Smith<br>Justin Lessler<br>Joseph C Lemairtre<br>Joshua Kaminsky<br>Alison Hill |
| University of Virginia | | Przemyslaw Porebski<br>Srinivasan Venkatramanan<br>Aniruddha Adiga<br>Bryan Lewis<br>Brian Klahn<br>Joseph Outten<br>Mark Orr<br>Galen Harrison<br>Benjamin Hurt<br>Jiangzhuo Chen<br>Anil Vullikanti<br>Madhav Marathe<br>Stefan Hoops<br>Parantapa Bhattacharya<br>Dustin Machi |

| Funder | Grant reference number | Author |
|---|---|---|
| COVID-19 HPC Consortium | | Przemyslaw Porebski<br>Srinivasan Venkatramanan<br>Aniruddha Adiga<br>Bryan Lewis<br>Brian Klahn<br>Joseph Outten<br>Mark Orr<br>Galen Harrison<br>Benjamin Hurt<br>Jiangzhuo Chen<br>Anil Vullikanti<br>Madhav Marathe<br>Stefan Hoops<br>Parantapa Bhattacharya<br>Dustin Machi |
| Amazon Web Services | | Shaun Truelove<br>Claire P Smith<br>Justin Lessler<br>Joseph C Lemairtre<br>Juan Dent<br>Joshua Kaminsky<br>Elizabeth C Lee<br>Javier Perez-Saez<br>Alison Hill |
| Models of Infectious Disease Agent Study | MIDASUP-05 | Shi Chen<br>Rajib Paul<br>Daniel Janies<br>Jean-Claude Thill |
| North Carolina Biotechnology Center | | Shi Chen<br>Rajib Paul<br>Daniel Janies<br>Jean-Claude Thill |
| National Institutes of Health | R01AI163023 | Marta Galanti<br>Teresa K Yamana<br>Sen Pei<br>Jeffrey L Shaman |
| Council of State and Territorial Epidemiologists | NU38OT000297 | Marta Galanti<br>Teresa K Yamana<br>Sen Pei<br>Jeffrey L Shaman |
| Morris-Singer Foundation | | Marta Galanti<br>Teresa K Yamana<br>Sen Pei<br>Jeffrey L Shaman |

The funders had no role in study design, data collection and interpretation, or the decision to submit the work for publication.

## Author contributions

Shaun Truelove, Conceptualization, Data curation, Formal analysis, Funding acquisition, Investigation, Methodology, Project administration, Software, Supervision, Validation, Visualization, Writing – original draft, Writing – review and editing; Claire P Smith, Conceptualization, Data curation, Formal analysis, Visualization, Writing – original draft, Writing – review and editing; Michelle Qin, Data curation, Formal analysis, Visualization, Writing – original draft, Writing – review and editing; Luke C Mullany, Conceptualization, Data curation, Formal analysis, Software, Visualization, Writing – original draft, Writing – review and editing; Rebecca K Borchering, Conceptualization, Formal analysis, Methodology, Project administration, Visualization, Writing – original draft, Writing – review and editing; Justin Lessler, Conceptualization, Funding acquisition, Methodology, Project administration, Supervision, Visualization, Writing – original draft, Writing – review and editing; Katriona Shea, Conceptualization, Funding acquisition, Methodology, Project administration, Supervision, Writing – original draft, Writing – review and editing; Emily Howerton, Formal analysis, Methodology, Software, Validation, Visualization, Writing – original draft, Writing – review and editing; Lucie Contamin, Formal analysis,

Methodology, Software, Visualization; John Levander, Lauren Shin, Brian Klahn, Joseph Outten, Jean-Claude Thill, Formal analysis, Methodology; Jessica Kerr, Data curation, Project administration, Resources, Visualization; Harry Hochheiser, Project administration, Resources, Software, Supervision, Writing – original draft, Writing – review and editing; Matt Kinsey, Kate Tallaksen, Joshua Kaminsky, Javier Perez-Saez, Kunpeng Mu, Xinyue Xiong, Ana Pastore y Piontti, Przemyslaw Porebski, Aniruddha Adiga, Mark Orr, Galen Harrison, Jiangzhuo Chen, Anil Vullikanti, Stefan Hoops, Parantapa Bhattacharya, Dustin Machi, Rajib Paul, Daniel Janies, Formal analysis, Methodology, Software; Shelby Wilson, Data curation, Methodology, Software; Kaitlin Rainwater-Lovett, Conceptualization, Funding acquisition, Methodology, Project administration, Supervision, Writing – review and editing; Joseph C Lemairtre, Juan Dent, Alison Hill, Matteo Chinazzi, Formal analysis, Methodology, Software, Writing – review and editing; Elizabeth C Lee, Conceptualization, Data curation, Formal analysis, Funding acquisition, Methodology, Software, Writing – review and editing; Dean Karlen, Jessica T Davis, Teresa K Yamana, Sen Pei, Formal analysis, Methodology, Writing – review and editing; Alessandro Vespignani, Bryan Lewis, Conceptualization, Formal analysis, Funding acquisition, Methodology, Project administration, Supervision, Writing – review and editing; Ajitesh Srivastava, Conceptualization, Formal analysis, Funding acquisition, Methodology, Writing – review and editing; Srinivasan Venkatramanan, Marta Galanti, Conceptualization, Formal analysis, Methodology, Writing – review and editing; Benjamin Hurt, Data curation, Formal analysis, Methodology; Madhav Marathe, Conceptualization, Formal analysis, Methodology, Supervision; Shi Chen, Formal analysis, Methodology, Project administration, Software; Jeffrey L Shaman, Formal analysis, Funding acquisition, Methodology, Software, Writing – review and editing; Jessica M Healy, Matthew Biggerstaff, Conceptualization, Funding acquisition, Project administration, Supervision, Writing – review and editing; Rachel B Slayton, Conceptualization, Funding acquisition, Project administration, Resources, Writing – review and editing; Michael A Johansson, Conceptualization, Funding acquisition, Project administration, Supervision; Michael C Runge, Conceptualization, Formal analysis, Methodology, Project administration, Supervision, Writing – original draft, Writing – review and editing; Cecile Viboud, Conceptualization, Formal analysis, Investigation, Methodology, Project administration, Supervision, Writing – original draft, Writing – review and editing

**Author ORCIDs**
Shaun Truelove  http://orcid.org/0000-0003-0538-0607
Rebecca K Borchering  http://orcid.org/0000-0003-4309-2913
Harry Hochheiser  http://orcid.org/0000-0001-8793-9982
Kaitlin Rainwater-Lovett  http://orcid.org/0000-0002-8707-7339
Juan Dent  http://orcid.org/0000-0003-3154-0731
Przemyslaw Porebski  http://orcid.org/0000-0001-8012-5791
Bryan Lewis  http://orcid.org/0000-0003-0793-6082
Jean-Claude Thill  http://orcid.org/0000-0002-6651-8123
Marta Galanti  http://orcid.org/0000-0002-9060-1250
Teresa K Yamana  http://orcid.org/0000-0001-8349-3151
Sen Pei  http://orcid.org/0000-0002-7072-2995
Jeffrey L Shaman  http://orcid.org/0000-0002-7216-7809
Michael C Runge  http://orcid.org/0000-0002-8081-536X
Cecile Viboud  http://orcid.org/0000-0003-3243-4711

**Decision letter and Author response**
Decision letter https://doi.org/10.7554/eLife.73584.sa1
Author response https://doi.org/10.7554/eLife.73584.sa2

---

## Additional files

**Supplementary files**
• Supplementary file 1. Summary of model assumptions for the seventh round of long-term scenario projections from the US COVID-19 Scenario Modeling Hub. This table details a summary of core model assumptions for the nine included models (arranged alphabetically).

• MDAR checklist

## Data availability

All model output data are available on the project github at https://github.com/midas-network/covid19-scenario-modeling-hub (archived at https://doi.org/10.5281/zenodo.6584489). Code and data specific to this manuscript has been consolidated into a repository at https://github.com/midas-network/covid19-scenario-modeling-hub/tree/master/paper-source-code/round-7. All data used are publicly available.

The following datasets were generated:

| Author(s) | Year | Dataset title | Dataset URL | Database and Identifier |
|---|---|---|---|---|
| Contamin L | 2022 | COVID-19 Scenario Modeling Hub | https://github.com/midas-network/covid19-scenario-modeling-hub | GitHub, midas-network/covid19-scenario-modeling-hub |
| Smith CP | 2022 | Projected resurgence of COVID-19 in the United States in July-December 2021 resulting from the increased transmissibility of the Delta variant and faltering vaccination | https://github.com/midas-network/covid19-scenario-modeling-hub/tree/master/paper-source-code/round-7 | GitHub, midas-network/covid19-scenario-modeling-hub/tree/master/paper-source-code/round-7 |

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
