## [Editor Report]

In this paper, the authors presented the joint efforts of nine modeling teams to provide a six-month projection of the COVID-19 pandemic across the US, in view of the circulation of the more transmissible Delta variant. The results represented a timely assessment of the risk of COVID-19 resurgence in Summer 2021 when it was conducted in July 2021, and will be of historical interest as an example of modeling efforts to inform real-time decision making during the COVID-19 pandemic. This paper will be of high interest to public health specialists, forecast modelers, and members of the general public interested in the evolution of the COVID-19 pandemic and the impact of public health interventions in the USA.

---

## [Decision Letter]

**Decision letter after peer review:**

Thank you for submitting your article "Projected resurgence of COVID-19 in the United States in July-December 2021 resulting from the increased transmissibility of the Δ variant and faltering vaccination" for consideration by *eLife*. Your article has been reviewed by 3 peer reviewers, one of whom is a member of our Board of Reviewing Editors, and the evaluation has been overseen by a Senior Editor. The following individual involved in review of your submission has agreed to reveal their identity: Jodie McVernon (Reviewer #3).

As is customary in *eLife*, the reviewers have discussed their critiques with one another. What follows below is the Reviewing Editor's edited compilation of the essential and ancillary points provided by reviewers in their critiques and in their interaction post-review. Please submit a revised version that addresses these concerns directly. Although we expect that you will address these comments in your response letter, we also need to see the corresponding revision clearly marked in the text of the manuscript. Some of the reviewers' comments may seem to be simple queries or challenges that do not prompt revisions to the text. Please keep in mind, however, that readers may have the same perspective as the reviewers. Therefore, it is essential that you attempt to amend or expand the text to clarify the narrative accordingly.

Essential revisions:

1. Provide a table describing the basic features of the models included in this analysis. Particular parameters that should be described include vaccine efficacy against infection and assumptions regarding natural immunity to reinfection.

2. Provide an analysis showing individual model predictions for a scenario in order to assess inter-model variability of predictions.

3. Provide further discussion regarding what assumptions of the models may have led to the underestimation of the number of cases in the Summer/Fall, and the features of the models that were more closely able to reproduce the observed resurgence.

4. Comment on the timeliness of predictions and the time horizon over which predictions are likely to be valid.

5. Discussion on whether heterogeneity below the state level could have contributed to the overestimation of vaccination effectiveness by the models.

6. Discussion on the impact of waning vaccine immunity, whether this was accounted for by the models, and whether this will be accounted for in future rounds of predictions.

*Reviewer #1 (Recommendations for the authors):*

– The current first section "overview" serves as both introduction, and summary of results. I think the manuscript would benefit from a more traditional introduction, where the objectives of the COVID-19 Scenario Modeling Hub are detailed, results from previous rounds are summarized, and the objectives of the current round (6? 7? I am not very clear on this) are detailed. The main results could be summarized in the discussion.

– There is very little information about the models which contributing predictions to this analysis. This does not allow the reader to fully assess the underlying methods. I would generally expect in any multiple model analysis to have a table describing basic features of the models, in the same way that a systematic review/meta-analysis would describe the features of the underlying studies. Key features to describe would be the type of model structure, design, and basic underlying assumptions likely to vary between models such as effectiveness against infection, transmission, and progression to severe outcomes, demographic stratifications included etc.

– The main value of combining different model predictions is to capture and explore uncertainty due to differences between models. Please consider including a figure of cross-model comparison of individual model predictions for a given scenario. Some discussion regarding differences between models leading to differences in predictions would also be warranted.

– I liked the comparison between model predictions and observed data for July. I would have liked to see some discussion relating to how this information is likely to influence future rounds of modelling and predictions, as it suggests that some of the assumptions of the models do not reflect the current epidemiology on the ground.

*Reviewer #2 (Recommendations for the authors):*

1) About the timeliness of the projections: I received the review invitation on 28 September 2021. Could the authors comment on the timeliness of projections? For example, if six-month projections were not possible, would it be more practical to provide projections every two months?

2) About model assumptions and Table 1: It is not clear to me what vaccine efficacy meant here in Table 1. Vaccine efficacy against infection, symptomatic infection, hospitalization, or death of COVID-19? Since the vaccine uptake by 3 July 2021 was the most important factor in the projections of cumulative cases and deaths in Figure 2, what were the assumptions in the nine models about immunity from vaccination and natural infections from the previous waves? Could the authors summarize the most important assumptions for each of the nine models?

3) About source of uncertainties: There were substantial uncertainties in the model projections in Figure 1 (i.e., the 95% CI of projections could cover nearly all the possible actual outcomes). Could the authors identify the major sources of uncertainties of each of the nine models?

Although all the data and codes are publicly available, it will be great if the authors could add a table to summarize the major similarities and differences of the nine models, given the high uncertainties in projections across different models in Figure 1.

*Reviewer #3 (Recommendations for the authors):*

This ensemble modelling exercise is incredibly useful, but could be even further enhanced by more of a discussion of difference, as indicated above. It will be very important to include more considerations of immune waning and boosting in future rounds.

Will future rounds incorporate assumptions about the duration of protection following the primary series, particularly as boosters come online and will also need to be incorporated? Could waning be a partial explanation for the overestimation of vaccine impacts here?

Naturally acquired immunity is mentioned but is natural boosting of immunised individuals built into any of these models?

---

## [Author Response]

Essential revisions:1. Provide a table describing the basic features of the models included in this analysis. Particular parameters that should be described include vaccine efficacy against infection and assumptions regarding natural immunity to reinfection.

We have added Supplementary File 1, a table that provides a summary of the assumptions for each of the nine contributing models. We have added the following sentence to the Methods (line 101, pg 6):

“For details on individual model assumptions, see Supplementary File 1.”

2. Provide an analysis showing individual model predictions for a scenario in order to assess inter-model variability of predictions.

We have added Figure4-supplemental figures 1-3, which display the individual model projections, demonstrating the inter-model variability of predictions. We have added a reference to these figures on line (230-231). Further, our new supplemental tables describing individual model assumptions will help readers better understand some of the drivers of inter-model variability (Supplementary File 1).

Although we comment on some of the drivers of this variability in the text (see 4th paragraph of Discussion, pg 11), we believe that a formal analysis of individual model predictions is beyond the scope of this paper. Differences between models are many and include variation in model structure, parameters, and calibration; this understanding the drivers of model variation is not straightforward. Overall, there are many dimensions of uncertainty in the epidemiologic situation, and it is likely that no individual model appropriately captures all these aspects of uncertainty. By combining the models together into an ensemble, we aim to better capture the range of this uncertainty and therefore to provide projections that are more useful for public health decision making.

3. Provide further discussion regarding what assumptions of the models may have led to the underestimation of the number of cases in the Summer/Fall, and the features of the models that were more closely able to reproduce the observed resurgence.

We have added additional text to further detail why projections underestimate the observed resurgences.

In the discussion (pg 10), we now state:

“Possible mechanisms driving the underestimation of the observed Summer 2021 resurgence may include inaccurate assumptions about the transmissibility and severity of Delta (including changes in serial interval or severity of infection relative to other variants), the effectiveness of the vaccines against infection and transmission, heterogeneity in vaccine coverage at the county level or below, waning of natural and vaccine-derived immunity, changes in testing practices, and the interaction of these factors with NPIs and behavior change.^13–15^ Assumptions regarding these factors were left to the discretion of the teams and therefore vary across models.”

No individual model was able to successfully reproduce the observed resurgences at the state-level. Those models that tracked with the observed resurgence at the national level for certain scenarios (particularly USC) did not consistently do so at the state-level.

4. Comment on the timeliness of predictions and the time horizon over which predictions are likely to be valid.

These should not be considered predictions, but rather projections of the epidemic trajectory under a predefined set of assumptions (the “scenario”). It is likely that no scenario will perfectly align with reality over the projection period and hence a simple comparison between observations and projections (as in forecast) is not satisfactory. As a case in point, the Omicron variant emerged at the end of our projection period and was not part of the scenarios that were designed in June 2021 (round 7) and are discussed in this paper.

We note that a full evaluation of our scenario projection efforts is underway and includes various stages of the pandemic (not just the Δ wave). We find that accuracy, measured by coverage or other scoring statistics, does not solely depend on time horizon, but also on the stage of the pandemic and on scenario design. In this round of projection particularly, coverage degrades rapidly over the first 9 weeks of the projection period but recovers in subsequent weeks (as can be seen in Figure 1). A full evaluation is beyond the scope of this paper, and in the meantime, we show a figure comparing state-level projections and observations over the first four weeks of projections (Figure 3).

As regards timeliness, our round 7 projections were made publicly available in mid-July 2021, as COVID19 incidence had just started rising, with the peak of the Δ wave occurring in the first week of September 2021. Hence our projections afforded reasonable time for action.

We have added two sentences in conclusion to highlight the timeliness of our projections (pg 13) and note the inherent difficulties in evaluating scenario-based projections and changes in accuracy by time horizon (pg 12). We also comment that projections are a valid tool that can be used to demonstrate the anticipated impact of changes to the epidemiologic situation – in this case, the relative importance of vaccine coverage and the introduction of a new variant (conclusion).

5. Discussion on whether heterogeneity below the state level could have contributed to the overestimation of vaccination effectiveness by the models.

Teams only submit projections at the state and national level, however many of the contributing models are county-level (Columbia, MOBS, UVA-adaptive, JHU APL). Heterogeneity in vaccination coverage at the county-level (or below) not captured by the models due either to spatial structure or lack of county-level data may very well have contributed to the underestimation of the observed resurgences.

We have added a comment to this effect in discussion pg 10.

6. Discussion on the impact of waning vaccine immunity, whether this was accounted for by the models, and whether this will be accounted for in future rounds of predictions.

Waning of both natural and vaccine-derived immunity was left to the discretion of the individual modeling teams. Only one of the 9 teams included waning immunity in their models. At the time of this round of projections, very little was known about the amount and characteristics of SARS-CoV-2 immunity waning. Subsequent SMH rounds do incorporate waning assumptions; these, along with all rounds of SMH projections, can be accessed on the SMH website (see round 13 in particular, https://covid19scenariomodelinghub.org).

We acknowledge this is now likely a limitation of our projections and have added the following to the Discussion section of this paper to discuss this:

“Critically, eight of the nine teams did not include waning of natural or vaccine-derived immunity, an assumption that we now know is incorrect.^16^ Absence of waning resulted in a much smaller population susceptible to infection with Delta, thus reducing the overall potential magnitude that could be projected by these models.”

“In addition, these scenarios do not specify considerations of reinfections or breakthrough infections with the Δ variant, the waning of existing immunity, changes in NPIs and behavior, or vaccination among children aged <12 years starting in November 2021, all of which were expected to be important drivers of dynamics in the subsequent months.”

Reviewer #1 (Recommendations for the authors):– The current first section "overview" serves as both introduction, and summary of results. I think the manuscript would benefit from a more traditional introduction, where the objectives of the COVID-19 Scenario Modeling Hub are detailed, results from previous rounds are summarized, and the objectives of the current round (6? 7? I am not very clear on this) are detailed. The main results could be summarized in the discussion.

We have modified the overview paragraph to have a bit more traditional feel, splitting it into paragraphs and adding the following paragraph about the COVID-19 Scenario Modeling Hub:

“Established in December 2020, the COVID-19 Scenario Modeling Hub is an effort to apply a multiple-model approach to produce six-month projections of the state and national trajectories of cases, hospitalizations, and deaths in the US under defined scenarios.^2^ Scenarios from projection rounds have focused on control measures, vaccination availability and uptake, and emerging variants.^3”^

We did leave a summary of the main findings in the “Overview”, as we felt this was appropriate for this short report, but we reduced this to a briefer summary to mirror a more traditional research article format as suggested. Additionally, to make it clear that this paper is focused on Round 7, we have removed all references to Round 6, focusing on Round 7 only.

– There is very little information about the models which contributing predictions to this analysis. This does not allow the reader to fully assess the underlying methods. I would generally expect in any multiple model analysis to have a table describing basic features of the models, in the same way that a systematic review/meta-analysis would describe the features of the underlying studies. Key features to describe would be the type of model structure, design, and basic underlying assumptions likely to vary between models such as effectiveness against infection, transmission, and progression to severe outcomes, demographic stratifications included etc.

We have added a supplemental table (Supplementary File 1) which provide an overview of individual model structure and assumptions.

– The main value of combining different model predictions is to capture and explore uncertainty due to differences between models. Please consider including a figure of cross-model comparison of individual model predictions for a given scenario. Some discussion regarding differences between models leading to differences in predictions would also be warranted.

We have added Figure 1-supplemental figures 1-3 which show the results of the nine individual models. Additionally, we have added a supplemental table (Supplementary File 1) which provide an overview of individual model structure and assumptions. We provide the following commentary in the Discussion section:

“Further, for a given scenario, there is notable variation among individual model projections with regards to both the timing and the magnitude of the resurgence (Figure 1-supplemental figure 1-3). Variation likely reflects differences in model structure, projected vaccine coverage, projected variant growth, and importance of seasonal effects.”

– I liked the comparison between model predictions and observed data for July. I would have liked to see some discussion relating to how this information is likely to influence future rounds of modelling and predictions, as it suggests that some of the assumptions of the models do not reflect the current epidemiology on the ground.

We update the parameters specified in the scenarios each round to align with the most up to date evidence. For example, we have incorporated waning into later rounds (we have just completed round 13 focused on waning). Further, in later rounds, we paid particular attention to estimates of VE specific to the Delta variant that became available later in the year.

Reviewer #2 (Recommendations for the authors):1) About the timeliness of the projections: I received the review invitation on 28 September 2021. Could the authors comment on the timeliness of projections? For example, if six-month projections were not possible, would it be more practical to provide projections every two months?

This is a good point and we now comment on the timeless of our projections (conclusion p 13):

“Projections of Delta resurgence presented in this paper were made publicly available in early July 2021 ^3^, two months ahead of the peak of the Delta wave, providing actionable results. There is a trade-off between releasing projections in a timely manner to guide decisions, and projection accuracy and uncertainty that improve with incorporation of recent information.”

We acknowledge there was a longer lag to turn this work into a publication.

We do acknowledge that as the epidemiological situation changes over the projection period, none of the scenarios will be expected to align with reality. However, the goal of these projections is not to predict the future but rather to demonstrate the anticipated impact of changes in vaccination coverage or the introduction of a more transmissible variant. While on occasion, we have provided predictions for shorter horizons (eg two closely spaced rounds of 3-month projections for Omicron, rounds 10 and 11, 17 days apart), these are more appropriate for emergency situations. Overall, we believe that a longer projection period better conveys the long-term impacts of changes in disease dynamics.

2) About model assumptions and Table 1: It is not clear to me what vaccine efficacy meant here in Table 1. Vaccine efficacy against infection, symptomatic infection, hospitalization, or death of COVID-19? Since the vaccine uptake by 3 July 2021 was the most important factor in the projections of cumulative cases and deaths in Figure 2, what were the assumptions in the nine models about immunity from vaccination and natural infections from the previous waves? Could the authors summarize the most important assumptions for each of the nine models?

The vaccine effectiveness specified in the scenarios is against symptomatic disease. We have added a note to table 1 clarifying this point. Further, we have added a supplemental table (Supplementary File 1) which provide an overview of individual model structure and assumptions.

3) About source of uncertainties: There were substantial uncertainties in the model projections in Figure 1 (i.e., the 95% CI of projections could cover nearly all the possible actual outcomes). Could the authors identify the major sources of uncertainties of each of the nine models?

We note in the text several key drivers of uncertainty:

“Uncertainty may arise from three main sources: specification of the scenarios (e.g., uncertainty in transmissibility); errors in the structure or assumptions of individual models given a specific scenario (e.g., variations in assumptions about vaccination uptake); and inaccurate calibration based on incomplete or biased data (e.g., reporting backlogs). None of the four scenarios considered here are likely to precisely reflect the future reality over a six-month period.^2^”

Uncertainty in the ensemble, which is obtained by a linear opinion pool approach, is driven both by differences between the trajectories of individual models, and uncertainty on the trajectory of individual models. We do feel that because sources of uncertainty are so varied, a discussion of model-specific results and uncertainty is beyond the scope of this paper. The reader can also refer to new tables in the supplement that describe the major assumptions from each model (Supplementary File 1).

Although all the data and codes are publicly available, it will be great if the authors could add a table to summarize the major similarities and differences of the nine models, given the high uncertainties in projections across different models in Figure 1.

We have added a table to the supplement with the main assumptions of each of the contributing models.

Reviewer #3 (Recommendations for the authors):This ensemble modelling exercise is incredibly useful, but could be even further enhanced by more of a discussion of difference, as indicated above. It will be very important to include more considerations of immune waning and boosting in future rounds.

We have strengthened our discussion of sources of model differences and provided a supplementary table of model assumptions. Subsequent SMH rounds have focused on different scenarios of waning and immune escape, which we now point to in the revised discussion. Results can be found at https://covid19scenariomodelinghub.org/index.html.

Will future rounds incorporate assumptions about the duration of protection following the primary series, particularly as boosters come online and will also need to be incorporated? Could waning be a partial explanation for the overestimation of vaccine impacts here?

This is certainly a possibility for future rounds. Round 8, 11 and round 13 incorporated waning immunity and future rounds may further refine this (round 13 is publicly available here https://covid19scenariomodelinghub.org/index.html). It is possible that improperly accounting for waning contributed to the underestimation of cases during the projection period, which we now mention.

Naturally acquired immunity is mentioned but is natural boosting of immunised individuals built into any of these models?

This was left to the discretion of the teams.